# SPHERICAL CNNS

**Taco S. Cohen**[*]
University of Amsterdam

**Mario Geiger**[*]
EPFL

**Jonas Köhler**[*]
University of Amsterdam

**Max Welling**
University of Amsterdam & CIFAR

## ABSTRACT

Convolutional Neural Networks (CNNs) have become the method of choice for learning problems involving 2D planar images. However, a number of problems of recent interest have created a demand for models that can analyze spherical images. Examples include omnidirectional vision for drones, robots, and autonomous cars, molecular regression problems, and global weather and climate modelling. A naive application of convolutional networks to a planar projection of the spherical signal is destined to fail, because the space-varying distortions introduced by such a projection will make translational weight sharing ineffective.

In this paper we introduce the building blocks for constructing spherical CNNs. We propose a definition for the spherical cross-correlation that is both expressive and rotation-equivariant. The spherical correlation satisfies a generalized Fourier theorem, which allows us to compute it efficiently using a generalized (non-commutative) Fast Fourier Transform (FFT) algorithm. We demonstrate the computational efficiency, numerical accuracy, and effectiveness of spherical CNNs applied to 3D model recognition and atomization energy regression.

## 1 INTRODUCTION

Convolutional networks are able to detect local patterns regardless of their position in the image. Like patterns in a planar image, patterns on the sphere can move around, but in this case the "move" is a 3D rotation instead of a translation. In analogy to the planar CNN, we would like to build a network that can detect patterns regardless of how they are rotated over the sphere.

As shown in Figure 1, there is no good way to use translational convolution or cross-correlation[1] to analyze spherical signals. The most obvious approach, then, is to change the definition of cross-correlation by replacing filter translations by rotations. Doing so, we run into a subtle but important difference between the plane and the sphere: whereas the space of moves for the plane (2D translations) is itself isomorphic to the plane, the space of moves for the sphere (3D rotations) is a different, *three-dimensional* manifold called $SO(3)$[2]. It follows that the result of a spherical correlation (the output feature map) is to be considered a signal on $SO(3)$, not a signal on the sphere, $S^2$. For this reason, we deploy $SO(3)$ group correlation in the higher layers of a spherical CNN (Cohen and Welling, 2016).

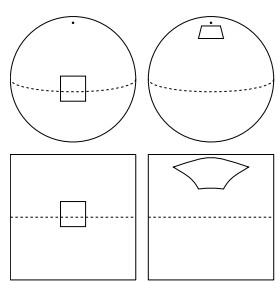

Figure 1: Any planar projection of a spherical signal will result in distortions. Rotation of a spherical signal cannot be emulated by translation of its planar projection.

---

[*]Equal contribution.

[1]Despite the name, CNNs typically use cross-correlation instead of convolution in the forward pass. In this paper we will generally use the term cross-correlation, or correlation for short.

[2]To be more precise: although the symmetry group of the plane contains more than just translations, the translations form a subgroup that acts on the plane. In the case of the sphere there is no coherent way to define a composition for points on the sphere, and so the sphere cannot act on itself (it is not a group). For this reason, we must consider the whole of $SO(3)$.

The implementation of a spherical CNN ($S^2$-CNN) involves two major challenges. Whereas a square grid of pixels has discrete translation symmetries, no perfectly symmetrical grids for the sphere exist. This means that there is no simple way to define the rotation of a spherical filter by one pixel. Instead, in order to rotate a filter we would need to perform some kind of interpolation. The other challenge is computational efficiency; SO(3) is a three-dimensional manifold, so a naive implementation of SO(3) correlation is $O(n^6)$.

We address both of these problems using techniques from non-commutative harmonic analysis (Chirikjian and Kyatkin, 2001; Folland, 1995). This field presents us with a far-reaching generalization of the Fourier transform, which is applicable to signals on the sphere as well as the rotation group. It is known that the SO(3) correlation satisfies a Fourier theorem with respect to the SO(3) Fourier transform, and the same is true for our definition of $S^2$ correlation. Hence, the $S^2$ and SO(3) correlation can be implemented efficiently using generalized FFT algorithms.

Because we are the first to use cross-correlation on a continuous group inside a multi-layer neural network, we rigorously evaluate the degree to which the mathematical properties predicted by the continuous theory hold in practice for our discretized implementation.

Furthermore, we demonstrate the utility of spherical CNNs for rotation invariant classification and regression problems by experiments on three datasets. First, we show that spherical CNNs are much better at rotation invariant classification of Spherical MNIST images than planar CNNs. Second, we use the CNN for classifying 3D shapes. In a third experiment we use the model for molecular energy regression, an important problem in computational chemistry.

### CONTRIBUTIONS

The main contributions of this work are the following:

1. The theory of spherical CNNs.

2. The first automatically differentiable implementation of the generalized Fourier transform for $S^2$ and SO(3). Our PyTorch code is easy to use, fast, and memory efficient.

3. The first empirical support for the utility of spherical CNNs for rotation-invariant learning problems.

## 2 RELATED WORK

It is well understood that the power of CNNs stems in large part from their ability to exploit (translational) symmetries though a combination of weight sharing and translation equivariance. It thus becomes natural to consider generalizations that exploit larger groups of symmetries, and indeed this has been the subject of several recent papers by Gens and Domingos (2014); Olah (2014); Dieleman et al. (2015; 2016); Cohen and Welling (2016); Ravanbakhsh et al. (2017); Zaheer et al. (2017b); Guttenberg et al. (2016); Cohen and Welling (2017). With the exception of SO(2)-steerable networks (Worrall et al., 2017; Weiler et al., 2017), these networks are all limited to discrete groups, such as discrete rotations acting on planar images or permutations acting on point clouds. Other very recent work is concerned with the analysis of spherical images, but does not define an equivariant architecture (Su and Grauman, 2017; Boomsma and Frellsen, 2017). Our work is the first to achieve equivariance to a continuous, non-commutative group (SO(3)), and the first to use the generalized Fourier transform for fast group correlation. A preliminary version of this work appeared as Cohen et al. (2017).

To efficiently perform cross-correlations on the sphere and rotation group, we use generalized FFT algorithms. Generalized Fourier analysis, sometimes called abstract- or noncommutative harmonic analysis, has a long history in mathematics and many books have been written on the subject (Sugiura, 1990; Taylor, 1986; Folland, 1995). For a good engineering-oriented treatment which covers generalized FFT algorithms, see (Chirikjian and Kyatkin, 2001). Other important works include (Driscoll and Healy, 1994; Healy et al., 2003; Potts et al., 1998; Kunis and Potts, 2003; Drake et al., 2008; Maslen, 1998; Rockmore, 2004; Kostelec and Rockmore, 2007; 2008; Potts et al., 2009; Makadia et al., 2007; Gutman et al., 2008).

## 3 CORRELATION ON THE SPHERE AND ROTATION GROUP

We will explain the $S^2$ and $SO(3)$ correlation by analogy to the classical planar $\mathbb{Z}^2$ correlation. The planar correlation can be understood as follows:

> The value of the output feature map at translation $x \in \mathbb{Z}^2$ is computed as an inner product between the input feature map and a filter, shifted by $x$.

Similarly, the spherical correlation can be understood as follows:

> The value of the output feature map evaluated at rotation $R \in SO(3)$ is computed as an inner product between the input feature map and a filter, rotated by $R$.

Because the output feature map is indexed by a rotation, it is modelled as a function on $SO(3)$. We will discuss this issue in more detail shortly.

The above definition refers to various concepts that we have not yet defined mathematically. In what follows, we will go through the required concepts one by one and provide a precise definition. Our goal for this section is only to present a mathematical model of spherical CNNs. Generalized Fourier theory and implementation details will be treated later.

**The Unit Sphere** $S^2$ can be defined as the set of points $x \in \mathbb{R}^3$ with norm 1. It is a two-dimensional manifold, which can be parameterized by spherical coordinates $\alpha \in [0, 2\pi]$ and $\beta \in [0, \pi]$.

**Spherical Signals** We model spherical images and filters as continuous functions $f : S^2 \to \mathbb{R}^K$, where $K$ is the number of channels.

**Rotations** The set of rotations in three dimensions is called $SO(3)$, the "special orthogonal group". Rotations can be represented by $3 \times 3$ matrices that preserve distance (i.e. $||Rx|| = ||x||$) and orientation ($\det(R) = +1$). If we represent points on the sphere as 3D unit vectors $x$, we can perform a rotation using the matrix-vector product $Rx$. The rotation group $SO(3)$ is a three-dimensional manifold, and can be parameterized by ZYZ-Euler angles $\alpha \in [0, 2\pi]$, $\beta \in [0, \pi]$, and $\gamma \in [0, 2\pi]$.

**Rotation of Spherical Signals** In order to define the spherical correlation, we need to know not only how to rotate points $x \in S^2$ but also how to rotate filters (i.e. functions) on the sphere. To this end, we introduce the rotation operator $L_R$ that takes a function $f$ and produces a rotated function $L_R f$ by composing $f$ with the rotation $R^{-1}$:

$$[L_R f](x) = f(R^{-1}x). \tag{1}$$

Due to the inverse on $R$, we have $L_{RR'} = L_R L_{R'}$.

**Inner products** The inner product on the vector space of spherical signals is defined as:

$$\langle \psi, f \rangle = \int_{S^2} \sum_{k=1}^{K} \psi_k(x) f_k(x) dx, \tag{2}$$

The integration measure $dx$ denotes the standard rotation invariant integration measure on the sphere, which can be expressed as $d\alpha \sin(\beta) d\beta / 4\pi$ in spherical coordinates (see Appendix A). The invariance of the measure ensures that $\int_{S^2} f(Rx) dx = \int_{S^2} f(x) dx$, for any rotation $R \in SO(3)$. That is, the volume under a spherical heightmap does not change when rotated. Using this fact, we can show that $L_{R^{-1}}$ is adjoint to $L_R$, which implies that $L_R$ is unitary:

$$\begin{aligned}
\langle L_R \psi, f \rangle &= \int_{S^2} \sum_{k=1}^{K} \psi_k(R^{-1}x) f_k(x) dx \\
&= \int_{S^2} \sum_{k=1}^{K} \psi_k(x) f_k(Rx) dx \\
&= \langle \psi, L_{R^{-1}} f \rangle.
\end{aligned} \tag{3}$$

**Spherical Correlation** With these ingredients in place, we are now ready to state mathematically what was stated in words before. For spherical signals $f$ and $\psi$, we define the correlation as:

$$[\psi \star f](R) = \langle L_R \psi, f \rangle = \int_{S^2} \sum_{k=1}^{K} \psi_k(R^{-1}x) f_k(x) dx. \tag{4}$$

As mentioned before, the output of the spherical correlation is a function on SO(3). This is perhaps somewhat counterintuitive, and indeed the conventional definition of spherical convolution gives as output a function on the sphere. However, as shown in Appendix B, the conventional definition effectively restricts the filter to be circularly symmetric about the Z axis, which would greatly limit the expressive capacity of the network.

**Rotation of** SO(3) **Signals** We defined the rotation operator $L_R$ for spherical signals (eq. 1), and used it to define spherical cross-correlation (eq. 4). To define the SO(3) correlation, we need to generalize the rotation operator so that it can act on signals defined on SO(3). As we will show, naively reusing eq. 1 is the way to go. That is, for $f : \text{SO}(3) \to \mathbb{R}^K$, and $R, Q \in \text{SO}(3)$:

$$[L_R f](Q) = f(R^{-1}Q). \tag{5}$$

Note that while the argument $R^{-1}x$ in Eq. 1 denotes the rotation of $x \in S^2$ by $R^{-1} \in \text{SO}(3)$, the analogous term $R^{-1}Q$ in Eq. 5 denotes to the composition of rotations (i.e. matrix multiplication).

**Rotation Group Correlation** Using the same analogy as before, we can define the correlation of two signals on the rotation group, $f, \psi : \text{SO}(3) \to \mathbb{R}^K$, as follows:

$$[\psi \star f](R) = \langle L_R \psi, f \rangle = \int_{\text{SO}(3)} \sum_{k=1}^K \psi_k(R^{-1}Q) f_k(Q) dQ. \tag{6}$$

The integration measure $dQ$ is the invariant measure on SO(3), which may be expressed in ZYZ-Euler angles as $d\alpha \sin(\beta) d\beta d\gamma/(8\pi^2)$ (see Appendix A).

**Equivariance** As we have seen, correlation is defined in terms of the rotation operator $L_R$. This operator acts naturally on the input space of the network, but what justification do we have for using it in the second layer and beyond?

The justification is provided by an important property, shared by all kinds of convolution and correlation, called equivariance. A layer $\Phi$ is equivariant if $\Phi \circ L_R = T_R \circ \Phi$, for some operator $T_R$. Using the definition of correlation and the unitarity of $L_R$, showing equivariance is a one liner:

$$[\psi \star [L_Q f]](R) = \langle L_R \psi, L_Q f \rangle = \langle L_{Q^{-1}R} \psi, f \rangle = [\psi \star f](Q^{-1}R) = [L_Q[\psi \star f]](R). \tag{7}$$

The derivation is valid for spherical correlation as well as rotation group correlation.

## 4 FAST SPHERICAL CORRELATION WITH G-FFT

It is well known that correlations and convolutions can be computed efficiently using the Fast Fourier Transform (FFT). This is a result of the Fourier theorem, which states that $\widehat{f * \psi} = \hat{f} \cdot \hat{\psi}$. Since the FFT can be computed in $O(n \log n)$ time and the product $\cdot$ has linear complexity, implementing the correlation using FFTs is asymptotically faster than the naive $O(n^2)$ spatial implementation.

For functions on the sphere and rotation group, there is an analogous transform, which we will refer to as the generalized Fourier transform (GFT) and a corresponding fast algorithm (GFFT). This transform finds it roots in the representation theory of groups, but due to space constraints we will not go into details here and instead refer the interested reader to Sugiura (1990) and Folland (1995).

Conceptually, the GFT is nothing more than the linear projection of a function onto a set of orthogonal basis functions called "matrix element of irreducible unitary representations". For the circle ($S^1$) or line ($\mathbb{R}$), these are the familiar complex exponentials $\exp(in\theta)$. For SO(3), we have the Wigner D-functions $D^l_{mn}(R)$ indexed by $l \geq 0$ and $-l \leq m, n \leq l$. For $S^2$, these are the spherical harmonics[3] $Y^l_m(x)$ indexed by $l \geq 0$ and $-l \leq m \leq l$.

Denoting the manifold ($S^2$ or SO(3)) by $X$ and the corresponding basis functions by $U^l$ (which is either vector-valued ($Y^l$) or matrix-valued ($D^l$)), we can write the GFT of a function $f : X \to \mathbb{R}$ as

$$\hat{f}^l = \int_X f(x) \overline{U^l(x)} dx. \tag{8}$$

---

[3]Technically, $S^2$ is not a group and therefore does not have irreducible representations, but it is a quotient of groups $\text{SO}(3)/\text{SO}(2)$ and we have the relation $Y^l_m = D^l_{m0}|_{S^2}$

This integral can be computed efficiently using a GFFT algorithm (see Section 4.1).

The inverse $SO(3)$ Fourier transform is defined as:

$$f(R) = \sum_{l=0}^{b}(2l+1)\sum_{m=-l}^{l}\sum_{n=-l}^{l}\hat{f}_{mn}^{l}D_{mn}^{l}(R), \tag{9}$$

and similarly for $S^2$. The maximum frequency $b$ is known as the bandwidth, and is related to the resolution of the spatial grid (Kostelec and Rockmore, 2007).

Using the well-known (in fact, defining) property of the Wigner D-functions that $D^l(R)D^l(R') = D^l(RR')$ and $D^l(R^{-1}) = D^l(R)^\dagger$, it can be shown (see Appendix D) that the $SO(3)$ correlation satisfies a Fourier theorem[4]: $\widehat{\psi \star f} = \hat{f} \cdot \hat{\psi}^\dagger$, where $\cdot$ denotes matrix multiplication of the two block matrices $\hat{f}$ and $\hat{\psi}^\dagger$.

Similarly, using $Y(Rx) = D(R)Y(x)$ and $Y_m^l = D_{m0}^l|_{S^2}$, one can derive an analogous $S^2$ convolution theorem: $\widehat{\psi \star f}^l = \hat{f}^l \cdot \hat{\psi}^{l\dagger}$, where $\hat{f}^l$ and $\hat{\psi}^l$ are now vectors. This says that the $SO(3)$-FT of the $S^2$ correlation of two spherical signals can be computed by taking the outer product of the $S^2$-FTs of the signals. This is shown in figure 2.

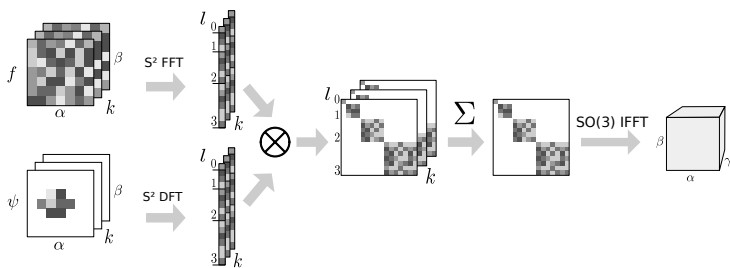

Figure 2: Spherical correlation in the spectrum. The signal $f$ and the locally-supported filter $\psi$ are Fourier transformed, block-wise tensored, summed over input channels, and finally inverse transformed. Note that because the filter is locally supported, it is faster to use a matrix multiplication (DFT) than an FFT algorithm for it. We parameterize the sphere using spherical coordinates $\alpha, \beta$, and $SO(3)$ with ZYZ-Euler angles $\alpha, \beta, \gamma$.

## 4.1 IMPLEMENTATION OF G-FFT AND SPECTRAL G-CONV

Here we sketch the implementation of GFFTs. For details, see (Kostelec and Rockmore, 2007).

The input of the $SO(3)$ FFT is a spatial signal $f$ on $SO(3)$, sampled on a discrete grid and stored as a 3D array. The axes correspond to the ZYZ-Euler angles $\alpha, \beta, \gamma$. The first step of the $SO(3)$-FFT is to perform a standard 2D translational FFT over the $\alpha$ and $\gamma$ axes. The FFT'ed axes correspond to the $m, n$ axes of the result. The second and last step is a linear contraction of the $\beta$ axis of the FFT'ed array with a precomputed array of samples from the Wigner-d (small-d) functions $d_{mn}^l(\beta)$. Because the shape of $d^l$ depends on $l$ (it is $(2l+1)\times(2l+1)$), this linear contraction is implemented as a custom GPU kernel. The output is a set of Fourier coefficients $\hat{f}_{mn}^l$ for $l \geq n, m \geq -l$ and $l = 0, \ldots, L_{\max}$.

The algorithm for the $S^2$-FFTs is very similar, only in this case we FFT over the $\alpha$ axis only, and do a linear contraction with precomputed Legendre functions over the $\beta$ axis.

Our code is available at `https://github.com/jonas-koehler/s2cnn`.

## 5 EXPERIMENTS

In a first sequence of experiments, we evaluate the numerical stability and accuracy of our algorithm. In a second sequence of experiments, we showcase that the new cross-correlation layers we have

---

[4]This result is valid for real functions. For complex functions, conjugate $\psi$ on the left hand side.

introduced are indeed useful building blocks for several real problems involving spherical signals. Our examples for this are recognition of 3D shapes and predicting the atomization energy of molecules.

## 5.1 EQUIVARIANCE ERROR

In this paper we have presented the first instance of a group equivariant CNN for a continuous, non-commutative group. In the discrete case, one can prove that the network is exactly equivariant, but although we can prove $[L_R f] * \psi = L_R[f * \psi]$ for continuous functions $f$ and $\psi$ on the sphere or rotation group, this is not exactly true for the discretized version that we actually compute. Hence, it is reasonable to ask if there are any significant discretization artifacts and whether they affect the equivariance properties of the network. If equivariance can not be maintained for many layers, one may expect the weight sharing scheme to become much less effective.

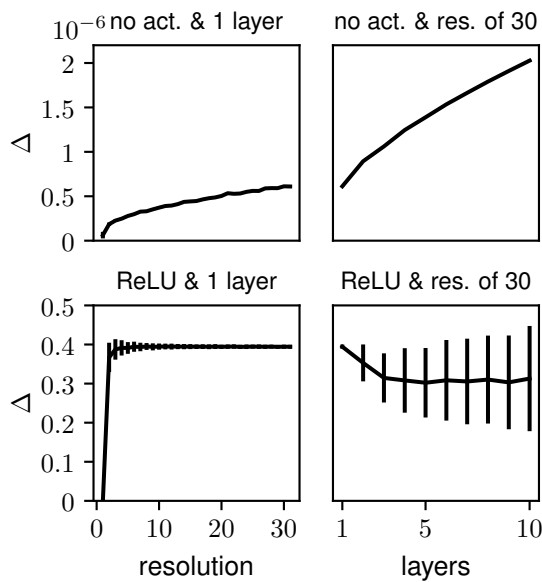

Figure 3: $\Delta$ as a function of the resolution and the number of layers.

We first tested the equivariance of the SO(3) correlation at various resolutions $b$. We do this by first sampling $n = 500$ random rotations $R_i$ as well as $n$ feature maps $f_i$ with $K = 10$ channels. Then we compute $\Delta = \frac{1}{n} \sum_{i=1}^{n} \mathrm{std}(L_{R_i} \Phi(f_i) - \Phi(L_{R_i} f_i)) / \mathrm{std}(\Phi(f_i))$, where $\Phi$ is a composition of SO(3) correlation layers with randomly initialized filters. In case of perfect equivariance, we expect this quantity to be zero. The results (figure 3 (top)), show that although the approximation error $\Delta$ grows with the resolution and the number of layers, it stays manageable for the range of resolutions of interest.

We repeat the experiment with ReLU activation function after each correlation operation. As shown in figure 3 (bottom), the error is higher but stays flat. This indicates that the error is not due to the network layers, but due to the feature map rotation, which is exact only for bandlimited functions.

## 5.2 ROTATED MNIST ON THE SPHERE

In this experiment we evaluate the generalization performance with respect to rotations of the input. For testing we propose a version MNIST dataset projected on the sphere (see fig. 4). We created two instances of this dataset: one in which each digit is projected on the northern hemisphere and one in which each projected digit is additionally randomly rotated.

**Architecture and Hyperparameters** As a baseline model, we use a simple CNN with layers conv-ReLU-conv-ReLU-FC-softmax, with filters of size $5 \times 5$, $k = 32, 64, 10$ channels, and stride 3 in both layers ($\approx 68$K parameters). We compare to a spherical CNN with layers $S^2$conv-ReLU-SO(3)conv-ReLU-FC-softmax, bandwidth $b = 30, 10, 6$ and $k = 20, 40, 10$ channels ($\approx 58$K parameters).

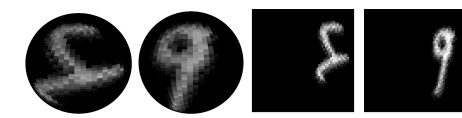

Figure 4: Two MNIST digits projected onto the sphere using stereographic projection. Mapping back to the plane results in non-linear distortions.

**Results** We trained each model on the non-rotated (NR) and the rotated (R) training set and evaluated it on the non-rotated and rotated test set. See table 1. While the planar CNN achieves high accuracy in the NR / NR regime, its performance in the R / R regime is much worse, while the spherical CNN is unaffected. When trained on the

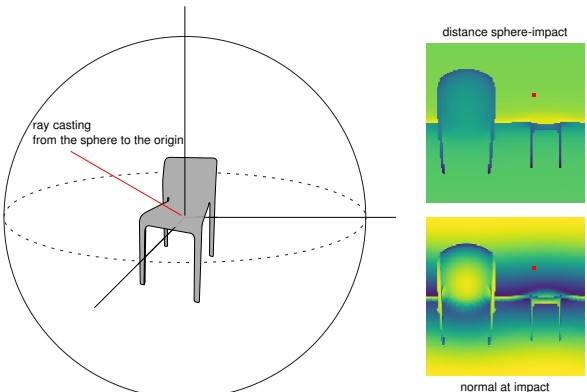

Figure 5: The ray is cast from the surface of the sphere towards the origin. The first intersection with the model gives the values of the signal. The two images of the right represent two spherical signals in $(\alpha, \beta)$ coordinates. They contain respectively the distance from the sphere and the cosine of the ray with the normal of the model. The red dot corresponds to the pixel set by the red line.

non-rotated dataset and evaluated on the rotated dataset (NR / R), the planar CNN does no better than random chance. The spherical CNN shows a slight decrease in performance compared to $R/R$, but still performs very well.

|  | NR / NR | R / R | NR / R |
|---|---|---|---|
| planar | 0.98 | 0.23 | 0.11 |
| spherical | 0.96 | 0.95 | 0.94 |

Table 1: Test accuracy for the networks evaluated on the spherical MNIST dataset. Here R = rotated, NR = non-rotated and X / Y denotes, that the network was trained on X and evaluated on Y.

## 5.3 RECOGNITION OF 3D SHAPES

Next, we applied $S^2$CNN to 3D shape classification. The SHREC17 task (Savva et al., 2017) contains 51300 3D models taken from the ShapeNet dataset (Chang et al., 2015) which have to be classified into 55 common categories (tables, airplanes, persons, etc.). There is a consistently aligned regular dataset and a version in which all models are randomly perturbed by rotations. We concentrate on the latter to test the quality of our rotation equivariant representations learned by $S^2$CNN.

**Representation**  We project the 3D meshes onto an enclosing sphere using a straightforward ray casting scheme (see Fig. 5). For each point on the sphere we send a ray towards the origin and collect 3 types of information from the intersection: ray length and $\cos / \sin$ of the surface angle. We further augment this information with ray casting information for the convex hull of the model, which in total gives us 6 channels for the signal. This signal is discretized using a Driscoll-Healy grid (Driscoll and Healy, 1994) with bandwidth $b = 128$. Ignoring non-convexity of surfaces we assume this projection captures enough information of the shape to be useful for the recognition task.

**Architecture and Hyperparameters**  Our network consists of an initial $S^2$conv-BN-ReLU block followed by two SO(3)conv-BN-ReLU blocks. The resulting filters are pooled using a max pooling layer followed by a last batch normalization and then fed into a linear layer for the final classification. It is important to note that the the max pooling happens over the group SO(3): if $f_k$ is the $k$-th filter in the final layer (a function on SO(3)) the result of the pooling is $\max_{x \in SO(3)} f_k(x)$. We used 50, 70, and 350 features for the $S^2$ and the two SO(3) layers, respectively. Further, in each layer we reduce the resolution $b$, from 128, 32, 22 to 7 in the final layer. Each filter kernel $\psi$ on SO(3) has non-local support, where $\psi(\alpha, \beta, \gamma) \neq 0$ iff $\beta = \frac{\pi}{2}$ and $\gamma = 0$ and the number of points of the discretization is proportional to the bandwidth in each layer. The final network contains $\approx 1.4$M parameters, takes 8GB of memory at batch size 16, and takes 50 hours to train.

| Method | P@N | R@N | F1@N | mAP | NDCG |
|--------|-----|-----|------|-----|------|
| Tatsuma_ReVGG | 0.705 | 0.769 | 0.719 | 0.696 | 0.783 |
| Furuya_DLAN | 0.814 | 0.683 | 0.706 | 0.656 | 0.754 |
| SHREC16-Bai_GIFT | 0.678 | 0.667 | 0.661 | 0.607 | 0.735 |
| Deng_CM-VGG5-6DB | 0.412 | 0.706 | 0.472 | 0.524 | 0.624 |
| **Ours** | 0.701 (3rd) | 0.711 (2nd) | 0.699 (3rd) | 0.676 (2nd) | 0.756 (2nd) |

Table 2: Results and best competing methods for the SHREC17 competition.

**Results**   We evaluated our trained model using the official metrics and compared to the top three competitors in each category (see table 2 for results). Except for precision and F1@N, in which our model ranks third, it is the runner up on each other metric. The main competitors, Tatsuma_ReVGG and Furuya_DLAN use input representations and network architectures that are highly specialized to the SHREC17 task. Given the rather task agnostic architecture of our model and the lossy input representation we use, we interpret our models performance as strong empirical support for the effectiveness of Spherical CNNs.

## 5.4   PREDICTION OF ATOMIZATION ENERGIES FROM MOLECULAR GEOMETRY

Finally, we apply $S^2$CNN on molecular energy regression. In the QM7 task (Blum and Reymond, 2009; Rupp et al., 2012) the atomization energy of molecules has to be predicted from geometry and charges. Molecules contain up to $N = 23$ atoms of $T = 5$ types (H, C, N, O, S). They are given as a list of positions $p_i$ and charges $z_i$ for each atom $i$.

**Representation by Coulomb matrices**   Rupp et al. (2012) propose a rotation and translation invariant representation of molecules by defining the *Coulomb matrix* $C \in \mathbb{R}^{N \times N}$ (CM). For each pair of atoms $i \neq j$ they set $C_{ij} = (z_i z_j)/(|p_i - p_j|)$ and $C_{ii} = 0.5 z_i^{2.4}$. Diagonal elements encode the atomic energy by nuclear charge, while other elements encode Coulomb repulsion between atoms. This representation is not permutation invariant. To this end Rupp et al. (2012) propose a distance measure between Coulomb matrices used within Gaussian kernels whereas Montavon et al. (2012) propose sorting $C$ or random sampling index permutations.

**Representation as a spherical signal**   We utilize spherical symmetries in the geometry by defining a sphere $S_i$ around around $p_i$ for each atom $i$. The radius is kept uniform across atoms and molecules and chosen minimal such that no intersections among spheres in the training set happen. Generalizing the Coulomb matrix approach we define for each possible $z$ and for each point $x$ on $S_i$ potential functions $U_z(x) = \sum_{j \neq i, z_j = z} \frac{z_i \cdot z}{|x - p_i|}$ producing a $T$ channel spherical signal for each atom in the molecule (see figure 6). This representation is invariant with respect to translations and equivariant with respect to rotations. However, it is still not permutation invariant. The signal is discretized using a Driscoll-Healy (Driscoll and Healy, 1994) grid with bandwidth $b = 10$ representing the molecule as a sparse $N \times T \times 2b \times 2b$ tensor.

**Architecture and Hyperparameters**   We use a deep ResNet style $S^2$CNN. Each ResNet block is made of $S^2$/SO(3)conv-BN-ReLU-SO(3)conv-BN after which the input is added to the result. We share weights among atoms making filters permutation invariant, by pushing the atom dimension into the batch dimension. In each layer we downsample the bandwidth, while increasing the number of features $F$. After integrating the signal over SO(3) each molecule becomes a $N \times F$ tensor. For permutation invariance over atoms we follow Zaheer et al. (2017a) and embed each resulting feature vector of an atom into a latent space using a MLP $\phi$. Then we sum these latent representations over the atom dimension and get our final regression value for the molecule by mapping with another MLP $\psi$. Both $\phi$ and $\psi$ are jointly optimized. Training a simple MLP only on the 5 frequencies of atom types in a molecule already gives a RMSE of $\sim 19$. Thus, we train the $S^2$CNN on the residual only, which improved convergence speed and stability over direct training. The final architecture is sketched in table 3. It has about 1.4M parameters, consumes 7GB of memory at batch size 20, and takes 3 hours to train.

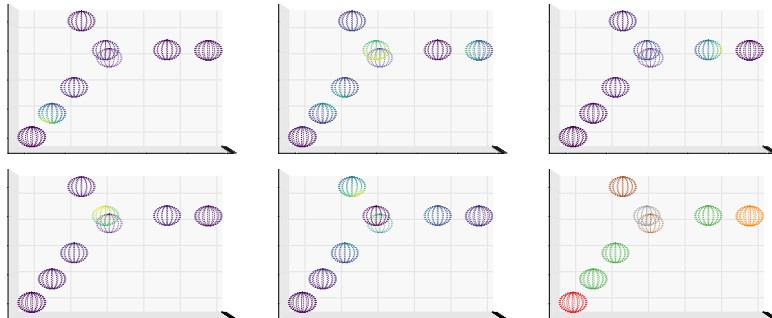

Figure 6: The five potential channels $U_z$ with $z \in \{1, 6, 7, 8, 16\}$ for a molecule containing atoms H (red), C (green), N (orange), O (brown), S (gray).

| Method | Author | RMSE | $S^2$CNN | Layer | Bandwidth | Features |
|---|---|---|---|---|---|---|
| MLP / random CM | (a) | 5.96 | | Input | | 5 |
| LGIKA(RF) | (b) | 10.82 | | ResBlock | 10 | 20 |
| RBF kernels / random CM | (a) | 11.40 | | ResBlock | 8 | 40 |
| RBF kernels / sorted CM | (a) | 12.59 | | ResBlock | 6 | 60 |
| MLP / sorted CM | (a) | 16.06 | | ResBlock | 4 | 80 |
| **Ours** | | **8.47** | | ResBlock | 2 | 160 |
| | | | DeepSet | Layer | Input/Hidden | |
| | | | | $\phi$ (MLP) | 160/150 | |
| | | | | $\psi$ (MLP) | 100/50 | |

Table 3: Left: Experiment results for the QM7 task: (a) Montavon et al. (2012) (b) Raj et al. (2016). Right: ResNet architecture for the molecule task.

**Results**  We evaluate by RMSE and compare our results to Montavon et al. (2012) and Raj et al. (2016) (see table 3). Our learned representation outperforms all kernel-based approaches and a MLP trained on sorted Coulomb matrices. Superior performance could only be achieved for an MLP trained on randomly permuted Coulomb matrices. However, sufficient sampling of random permutations grows exponentially with $N$, so this method is unlikely to scale to large molecules.

## 6 DISCUSSION & CONCLUSION

In this paper we have presented the theory of Spherical CNNs and evaluated them on two important learning problems. We have defined $S^2$ and SO(3) cross-correlations, analyzed their properties, and implemented a Generalized FFT-based correlation algorithm. Our numerical results confirm the stability and accuracy of this algorithm, even for deep networks. Furthermore, we have shown that Spherical CNNs can effectively generalize across rotations, and achieve near state-of-the-art results on competitive 3D Model Recognition and Molecular Energy Regression challenges, without excessive feature engineering and task-tuning.

For intrinsically volumetric tasks like 3D model recognition, we believe that further improvements can be attained by generalizing further beyond SO(3) to the roto-translation group SE(3). The development of Spherical CNNs is an important first step in this direction. Another interesting generalization is the development of a Steerable CNN for the sphere (Cohen and Welling, 2017), which would make it possible to analyze vector fields such as global wind directions, as well as other sections of vector bundles over the sphere.

Perhaps the most exciting future application of the Spherical CNN is in omnidirectional vision. Although very little omnidirectional image data is currently available in public repositories, the increasing prevalence of omnidirectional sensors in drones, robots, and autonomous cars makes this a very compelling application of our work.

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

APPENDIX A: PARAMETERIZATION OF AND INTEGRATION ON $S^2$ AND $SO(3)$

We use the ZYZ Euler parameterization for SO(3). An element $R \in SO(3)$ is written as

$$R = R(\alpha, \beta, \gamma) = Z(\alpha)Y(\beta)Z(\gamma), \tag{10}$$

where $\alpha \in [0, 2\pi]$, $\beta \in [0, \pi]$ and $\gamma \in [0, 2\pi]$, and $Z$ resp. $Y$ are rotations around the Z and Y axes.

Using this parameterization, the normalized Haar measure is

$$dR = \frac{d\alpha}{2\pi} \frac{d\beta \sin(\beta)}{2} \frac{d\gamma}{2\pi} \tag{11}$$

We have $\int_{SO(3)} dR = 1$. The Haar measure (Nachbin, 1965; Chirikjian and Kyatkin, 2001) is sometimes called the invariant measure because it has the property that $\int_{SO(3)} f(R'R)dR = \int_{SO(3)} f(R)dR$ (this is analogous to the more familiar property $\int_{\mathbb{R}} f(x + y)dx = \int_{\mathbb{R}} f(x)dx$ for functions on the line). This invariance property allows us to do many useful substitutions.

We have a related parameterization for the sphere. An element $x \in S^2$ is written

$$x(\alpha, \beta) = Z(\alpha)Y(\beta)n \tag{12}$$

where $n$ is the north pole.

This parameterization makes explicit the fact that the sphere is a quotient $S^2 = SO(3)/SO(2)$, where $H = SO(2)$ is the subgroup of rotations around the Z axis. Elements of this subgroup $H$ leave the north pole invariant, and have the form $Z(\gamma)$. The point $x(\alpha, \beta) \in S^2$ is associated with the coset representative $\bar{x} = R(\alpha, \beta, 0) \in SO(3)$. This element represents the coset $\bar{x}H = \{R(\alpha, \beta, \gamma) | \gamma \in [0, 2\pi]\}$.

The normalized Haar measure for the sphere is

$$dx = \frac{d\alpha}{2\pi} \frac{d\beta \sin \beta}{2} \tag{13}$$

The normalized Haar measure for SO(2) is

$$dh = \frac{d\gamma}{2\pi} \tag{14}$$

So we have $dR = dx\, dh$, again reflecting the quotient structure.

We can think of a function on $S^2$ as a $\gamma$-invariant function on SO(3). Given a function $f : S^2 \to \mathbb{C}$ we associate the function $\bar{f}(\alpha, \beta, \gamma) = f(\alpha, \beta)$. When using normalized Haar measures, we have:

$$
\begin{aligned}
\int_{SO(3)} \bar{f}(R)dR &= \frac{1}{8\pi^2} \int_0^{2\pi} d\alpha \int_0^\pi \sin\beta d\beta \int_0^{2\pi} d\gamma \bar{f}(\alpha, \beta, \gamma) \\
&= \frac{1}{8\pi^2} \int_0^{2\pi} d\alpha \int_0^\pi \sin\beta d\beta f(\alpha, \beta) \int_0^{2\pi} d\gamma \\
&= \frac{1}{4\pi} \int_0^{2\pi} d\alpha \int_0^\pi \sin\beta d\beta f(\alpha, \beta) \\
&= \int_{S^2} f(x)dx.
\end{aligned}
\tag{15}
$$

This will allow us to define the Fourier transform on $S^2$ from the Fourier transform on SO(3), by viewing a function on $S^2$ as a $\gamma$-invariant function on SO(3) and taking its SO(3)-Fourier transform.

APPENDIX B: CORRELATION & EQUIVARIANCE

We have defined the $S^2$ correlation as

$$[\psi \star f](R) = \langle L_R \psi, f \rangle = \int_{S^2} \sum_{k=1}^K \psi_k(R^{-1}x) f_k(x)dx. \tag{16}$$

Without loss of generality, we will analyze here the single-channel case $K = 1$.

This operation is equivariant:

$$
\begin{aligned}
[\psi \star [L_Q f]](R) &= \int_{S^2} \psi(R^{-1}x) f(Q^{-1}x) dx \\
&= \int_{S^2} \psi(R^{-1}Qx) f(x) dx \\
&= \int_{S^2} \psi((Q^{-1}R)^{-1}x) f(x) dx \\
&= [\psi \star f](Q^{-1}R) \\
&= [L_Q[\psi \star f]](R)
\end{aligned}
\tag{17}
$$

A similar derivation can be made for the SO(3) correlation.

The spherical convolution defined by Driscoll and Healy (1994) is:

$$
[f * \psi](x) = \int_{\mathrm{SO}(3)} f(Rn)\psi(R^{-1}x) dR
\tag{18}
$$

where $n$ is the north pole. Note that in this definition, the output of the spherical convolution is a function on the sphere, not a function on SO(3) as in our definition of cross-correlation. Note further that unlike our definition, this definition involves an integral over SO(3).

If we write out the integral in terms of Euler angles, noting that the north-pole $n$ is invariant to $Z$-axis rotations by $\gamma$, i.e. $R(\alpha, \beta, \gamma)n = Z(\alpha)Y(\beta)Z(\gamma)n = Z(\alpha)Y(\beta)n$, we see that this definition implicitly integrates over $\gamma$ in only one of the factors (namely $\psi$), making it invariant wrt $\gamma$ rotation. In other words, the filter is first "averaged" (making it circularly symmetric) before it is combined with $f$ (This was observed before by Makadia et al. (2007)). We consider this to be much too limited for the purpose of pattern matching in spherical CNNs.

## APPENDIX C: GENERALIZED FOURIER TRANSFORM

With each compact topological group (like SO(3)) is associated a discrete set of orthogonal functions that arise as matrix elements of irreducible unitary representations of these groups. For the circle (the group SO(2)) these are the complex exponentials (in the complex case) or sinusoids (for real functions). For SO(3), these functions are known as the Wigner D-functions.

As discussed in the paper, the Wigner D-functions are parameterized by a degree parameter $l \geq 0$ and order parameters $m, n \in [-l, \ldots, l]$. In other words, we have a set of matrix-valued functions $D^l : \mathrm{SO}(3) \to \mathbb{C}^{(2l+1) \times (2l+1)}$.

The Wigner D-functions are orthogonal:

$$
\langle D^l_{mn}, D^{l'}_{m'n'} \rangle = \int_0^{2\pi} \frac{d\alpha}{2\pi} \int_0^\pi \frac{d\beta \sin\beta}{2} \int_0^{2\pi} \frac{d\gamma}{2\pi} D^l_{mn}(\alpha, \beta, \gamma) \overline{D^{l'}_{m'n'}}(\alpha, \beta, \gamma) = \frac{\delta_{ll'}\delta_{mm'}\delta_{nn'}}{2l+1}
\tag{19}
$$

Furthermore, they are complete, meaning that any well behaved function $f : \mathrm{SO}(3) \to \mathbb{C}$ can be written as a linear combination of Wigner D-functions. This is the idea of the Generalized Fourier Transform $\mathcal{F}$ on SO(3):

$$
f(R) = [\mathcal{F}^{-1}\hat{f}](R) = \sum_{l=0}^\infty (2l+1) \sum_{m=-l}^l \sum_{n=-l}^l \hat{f}^l_{mn} D^l_{mn}(R)
\tag{20}
$$

where $\hat{f}^l_{mn}$ are called the Fourier coefficients of $f$. Using the orthogonality property of the Wigner D-functions, one can see that the Fourier coefficients can be retrieved by computing the inner product with the Wigner D-functions:

$$
\begin{aligned}
[\mathcal{F}f]^l_{mn} &= \int_{\mathrm{SO}(3)} f(R)\overline{D^l_{mn}(R)}dR \\
&= \int_{\mathrm{SO}(3)} \left[ \sum_{l'=0}^{\infty}(2l'+1)\sum_{m'=-l'}^{l'}\sum_{n'=-l'}^{l'} \hat{f}^{l'}_{m'n'}D^{l'}_{m'n'}(R) \right] \overline{D^l_{mn}(R)}dR \\
&= \sum_{l'=0}^{\infty}(2l'+1)\sum_{m'=-l'}^{l'}\sum_{n'=-l'}^{l'} \hat{f}^{l'}_{m'n'}\int_{\mathrm{SO}(3)} D^{l'}_{m'n'}(R)\overline{D^l_{mn}}dR \\
&= \hat{f}^l_{mn}
\end{aligned}
\tag{21}
$$

## APPENDIX D: FOURIER THEOREMS

Fourier convolution theorems for $\mathrm{SO}(3)$ and $\S^2$ can be found in Kostelec and Rockmore (2008); Makadia et al. (2007); Gutman et al. (2008). We derive them here for completeness.

To derive the convolution theorems, we will use the defining property of the Wigner D-matrices: that they are (irreducible, unitary) *representations* of $\mathrm{SO}(3)$. This means that they satisfy:

$$
D^l(R)D^l(R') = D^l(RR'),
\tag{22}
$$

for any $R, R' \in \mathrm{SO}(3)$. Notice that the complex exponentials satisfy an analogous criterion for the circle group $S^1 \cong \mathrm{SO}(2)$. That is, $e^{inx}e^{iny} = e^{in(x+y)}$, where $x + y$ is the group operation for $\mathrm{SO}(2)$.

Unitarity means that $D^l(R)D^{l\dagger}(R) = I$. Irreducibility means, essentially, that the set of matrices $\{D^l(R)\,|\,R \in \mathrm{SO}(3)\}$ cannot be simultaneously block-diagonalized.

To derive the Fourier theorem for $\mathrm{SO}(3)$, we use the invariance of the integration measure $dR$: $\int_{\mathrm{SO}(3)} f(R'R)dR = \int_{\mathrm{SO}(3)} f(R)dR$.

With these facts understood, we can proceed to derive:

$$
\begin{aligned}
\widehat{\psi \star f}^l &= \int_{\mathrm{SO}(3)} (\psi \star f)(R)\overline{D^l(R)}dR \\
&= \int_{\mathrm{SO}(3)} \int_{\mathrm{SO}(3)} \psi(R^{-1}R')f(R')dR'\overline{D^l(R)}dR \\
&= \int_{\mathrm{SO}(3)} \int_{\mathrm{SO}(3)} \psi(R^{-1})f(R')\overline{D^l(R'R)}dR'\,dR \\
&= \int_{\mathrm{SO}(3)} f(R')\overline{D^l(R')}dR' \int_{\mathrm{SO}(3)} \psi(R^{-1})\overline{D^l(R)}dR \\
&= \int_{\mathrm{SO}(3)} f(R')\overline{D^l(R')}dR' \int_{\mathrm{SO}(3)} \psi(R)\overline{D^l(R)}^{\dagger}dR \\
&= \hat{f}^l\,\hat{\psi}^{l\dagger}
\end{aligned}
\tag{23}
$$

So the $\mathrm{SO}(3)$-Fourier transform of the $\mathrm{SO}(3)$ convolution of $\psi$ and $f$ is equal to the matrix product of the $\mathrm{SO}(3)$-Fourier transforms $\hat{f}$ and $\hat{\psi}$.

For the sphere, we can derive an analogous transform that is sometimes called the spherical harmonics transform. The spherical harmonics $Y^l_m : S^2 \to \mathbb{C}$ are a complete orthogonal family of functions. The spherical harmonics are related to the Wigner D functions by the relation $D^l_{mn}(\alpha, \beta, \gamma) = Y^l_m(\alpha, \beta)e^{in\gamma}$, so that $Y^l_m(\alpha, \beta) = D^l_{m0}(\alpha, \beta, 0)$.

The $S^2$ convolution of $f_1$ and $f_2$ is equivalent to the SO(3) convolution of the associated right-invariant functions $\bar{f}_1, \bar{f}_2$ (see Appendix A):

$$
\begin{aligned}
[f_1 \star f_2](R) &= \int_{S^2} f_1(R^{-1}x) f_2(x) dx \\
&= \int_{SO(2)} \int_{S^2} f_1(R^{-1}x) f_2(x) dx dh \\
&= \int_{SO(3)} \bar{f}_1(R^{-1}R') \bar{f}_2(R') dR' \\
&= [\bar{f}_1 \star \bar{f}_2](R)
\end{aligned}
\tag{24}
$$

The Fourier transform of a right invariant function on SO(3) equals

$$
\begin{aligned}
[\mathcal{F}\bar{f}]_{mn}^l &= \int_0^{2\pi} \frac{d\alpha}{2\pi} \int_0^{\pi} \frac{d\beta \sin\beta}{2} \int_0^{2\pi} \frac{d\gamma}{2\pi} \bar{f}(\alpha, \beta, \gamma) \overline{D_{mn}^l(\alpha, \beta, \gamma)} \\
&= \int_0^{2\pi} \frac{d\alpha}{2\pi} \int_0^{\pi} \frac{d\beta \sin\beta}{2} f(\alpha, \beta) \int_0^{2\pi} \frac{d\gamma}{2\pi} \overline{D_{mn}^l(\alpha, \beta, \gamma)} \\
&= \delta_{n0} \int_0^{2\pi} \frac{d\alpha}{2\pi} \int_0^{\pi} \frac{d\beta \sin\beta}{2} f(\alpha, \beta) \overline{D_{m0}^l(\alpha, \beta, 0)} \\
&= \delta_{n0} \int_{S^2} f(x) \overline{Y_m^l(x)} dx
\end{aligned}
\tag{25}
$$

So we can think of the $S^2$ Fourier transform of a function on $S^2$ as the $n = 0$ column of the SO(3) Fourier transform of the associated right-invariant function. This is a beautiful result that we have not been able to find a reference for, though it seems likely that it has been observed before.

