# OpenReview forum: "Spherical CNNs"
_ICLR.cc/2018/Conference — Accept (Oral)_

### Official Review · AnonReviewer1 · 2017-11-27
**Spherical CNNs**

**Rating:** 8
**Confidence:** 4

**Review:**

Summary:

The paper proposes a framework for constructing spherical convolutional networks (ConvNets) based on a novel synthesis of several existing concepts.  The goal is to detect patterns in spherical signals irrespective of how they are rotated on the sphere.  The key is to make the convolutional architecture rotation equivariant.

Pros:

+ novel/original proposal justified both theoretically and empirically
+ well written, easy to follow
+ limited evaluation on a classification and regression task is suggestive of the proposed approach's potential
+ efficient implementation

Cons:

- related work, in particular the first paragraph, should compare and contrast with the closest extant work rather than merely list them
- evaluation is limited; granted this is the nature of the target domain

Presentation:

While the paper is generally written well, the paper appears to conflate the definition of the convolutional and correlation operators?  This point should be clarified in a revised manuscript.

In Section 5 (Experiments), there are several references to S^2CNN.  This naming of the proposed approach should be made clear earlier in the manuscript.  As an aside, this appears a little confusing since convolution is performed first on S^2 and then SO(3).

Evaluation:

What are the timings of the forward/backward pass and space considerations for the Spherical ConvNets presented in the evaluation section?  Please provide specific numbers for the various tasks presented.

How many layers (parameters) are used in the baselines in Table 2?  If indeed there are much less parameters used in the proposed approach, this would strengthen the argument for the approach.  On the other hand, was there an attempt to add additional layers to the proposed approach for the shape recognition experiment in Sec. 5.3 to improve performance?

Minor Points:

- some references are missing their source, e.g., Maslen 1998 and Kostolec, Rockmore, 2007, and Ravanbakhsh, et al. 2016.

- some sources for the references are presented inconsistency, e.g., Cohen and Welling, 2017 and Dieleman, et al. 2017

- some references include the first name of the authors, others use the initial

- in references to et al. or not, appears inconsistent

- Eqns 4, 5, 6, and 8 require punctuation

- Section 4 line 2, period missing before "Since the FFT"

- "coulomb matrix" --> "Coulomb matrix"

- Figure 5, caption: "The red dot correcpond to" --> "The red dot corresponds to"

Final remarks:

Based on the novelty of the approach, and the sufficient evaluation, I recommend the paper be accepted.

---

> ### Author Response · Authors · 2018-01-02
> **Spherical CNNs**
>
> Thank you for the detailed and balanced review.
>
> RE Related work: we have expanded the related work section a little bit in order to contrast with previous work. (Unfortunately there is no space for a very long discussion)
>
> RE Convolution vs correlation: thank you for pointing this out. Our reasoning had been that:
> 1) Everybody in deep learning uses the word "convolution" to mean "cross-correlation".
> 2) In the non-commutative case, there are several different but essentially equivalent convolution-like integrals that one can define, with no really good reason to prefer one over the other.
>
> But we did not explain this properly. We think a reasonable approach is to call something group convolution if, for the translation group it specializes to the standard convolution, and similarly for group correlations. This seems to be what several others before us have done as well, so we will follow this convention. Specifically, we will define the (group) cross-correlation as:
>   psi \star f(g) = int psi(g^{-1} h) f(h) dh.
>
> RE The S^2CNN name: we have now defined this term in the introduction, but not changed it, because the paper is called "Spherical CNN" and S^2-CNN is just a shorthand for that name.
>
> RE Timings: we have added timings, memory usage numbers, and number of parameters to the paper. It is not always possible to compare the number of parameters to related work because those numbers are not always available. However, we can reasonably assume that the competing methods did their own cross-validation to arrive at an optimal model complexity for their architecture. (Also, in deep networks, the absolute number of parameters can often vary widely between architectures that have a similar generalization performance, making this a rather poor measure of model complexity.)
>
> RE References and other minor points: we have fixed all of these issues. Thanks for pointing them out.

---

### Official Review · AnonReviewer3 · 2017-11-27
**Non-Abelian Harmonic Analysis to Get Spherical Invariance in CNNs**

**Rating:** 7
**Confidence:** 3

**Review:**

The focus of the paper is how to extend convolutional neural networks to have built-in spherical invariance.  Such a requirement naturally emerges when working with omnidirectional vision (autonomous cars, drones, ...).

To get invariance on the sphere (S^2), the idea is to consider the group of rotations on S^2 [SO(3)] and spherical convolution [Eq. (4)]. To be able to compute this convolution efficiently, a generalized Fourier theorem is useful. In order to achieve this goal, the authors adapt tools from non-Abelian [SO(3)] harmonic analysis.  The validity of the idea is illustrated on 3D shape recognition and atomization energy prediction.

The paper is nicely organized and clearly written; it fits to the focus of ICLR and can be applicable on many other domains as well.

---

> ### Author Response · Authors · 2018-01-02
> **Thanks**
>
> Thank you very much for taking the time to review our work.

---

> > ### Comment · AnonReviewer3 · 2018-01-12
> > **Thanks**
> >
> > Thank you for the feedback; I maintain my opinion.

---

### Official Review · AnonReviewer2 · 2017-12-11
**Added Late Reviewer**

**Rating:** 9
**Confidence:** 4

**Review:**

First off, this paper was a delight to read.  The authors develop an (actually) novel scheme for representing spherical data from the ground up, and test it on three wildly different empirical tasks: Spherical MNIST, 3D-object recognition, and atomization energies from molecular geometries.  They achieve near state-of-the-art performance against other special-purpose networks that aren't nearly as general as their new framework.  The paper was also exceptionally clear and well written.

The only con (which is more a suggestion than anything)--it would be nice if the authors compared the training time/# of parameters of their model versus the closest competitors for the latter two empirical examples.  This can sometimes be an apples-to-oranges comparison, but it's nice to fully contextualize the comparative advantage of this new scheme over others.  That is, does it perform as well and train just as fast?  Does it need fewer parameters?  etc.

I strongly endorse acceptance.

---

> ### Author Response · Authors · 2018-01-02
> **Spherical CNNs**
>
> Thank you for the kind words, we're glad you like our work!
>
> Our models for SHREC17 and QM7 both use only about 1.4M parameters. On a machine with 1 Titan X GPU, training the SHREC17 model takes about 50 hours, while the QM7 model takes only about 3 hours. Memory usage is 8GB for SHREC (batchsize 16) and 7GB for QM7 (batchsize 20).
>
> We have studied the SHREC17 paper [1], but unfortunately it does not state the number of parameters or training time for the various methods. It does seem likely that each of the competition participants did their own cross validation, and arrived at an appropriate model complexity for their method. It is thus unlikely that the strong performance of our model relative to others can be explained by its size (especially since 1.4M parameters is not considered very large anymore).
>
> For QM7, it looks like Montavon et al. used about 760k parameters (we have deduced this from the description of their network architecture). Since the model is a simple multi-layer perceptron applied to a hand-designed feature representation, we expect that it is substantially faster to train than our model (though indeed comparing a spherical CNN to an engineered features+MLP approach is a bit of an apples-to-oranges comparison). Raj et al. use a non-parametric method, so there is no parameter count or training time to compare to.
>
> [1] M. Savva et al. SHREC’17 Track Large-Scale 3D Shape Retrieval from ShapeNet Core55, Eurographics Workshop on 3D Object Retreival (2017).

---

### Public Comment · (anonymous) · 2017-12-19
**Spherical correlation, not convolution**

   In page 5: "This says that the SO(3)-FT of the S2 convolution (as we have defined it) of two spherical signals can  be computed by taking the outer product of the S2-FTs of the signals. This is shown in figure 2. We were unable to find a reference for the latter version of the S2 Fourier theorem"

   The result is presented at least in:
   - Makadia et al. (2007), eq (21),
   - Kostelec and Rockmore (2008), eq (6.6),
   - Gutman et al. (2008), eq (9),
   - Rafaely (2015), eq (1.88).

   All mentioned references define "spherical correlation" as what you define as "spherical convolution". I believe it makes more sense to call it correlation, since it can be seen as a measure of similarity between two functions (given two functions on S2 and transformations on SO(3), the correlation function measures the similarity as a function of the transformation).

   References:
   Makadia, A., Geyer, C., & Daniilidis, K., Correspondence-free structure from motion, International Journal of Computer Vision, 75(3), 311–327 (2007).
   Kostelec, P. J., & Rockmore, D. N., Ffts on the rotation group, Journal of Fourier analysis and applications, 14(2), 145–179 (2008).
   Gutman, B., Wang, Y., Chan, T., Thompson, P. M., & Toga, A. W., Shape registration with spherical cross correlation, 2nd MICCAI workshop on mathematical foundations of computational anatomy (pp. 56–67) (2008).
   Rafaely B. Fundamentals of spherical array processing. Berlin: Springer; (2015).

---

> ### Author Response · Authors · 2018-01-02
> **Thanks**
>
> Thank you for these references, they are indeed very relevant and interesting*. We will add them and change the text.
>
> We agree that the cross-correlation is the right term, and have fixed it in the paper. We have added further discussion of this issue in reply to reviewer 2, who raised a similar concern.
>
> * We do not have access to Rafaely's book through our university library, so we cannot comment on it.

---

### Public Comment · ~Tao_Sun1 · 2018-01-26
**Relationship with "Convolutional Networks for Spherical Signals"**

How to describe the relationships between these two papers?

---

> ### Author Response · Authors · 2018-01-30
> **Workshop paper**
>
> The paper [1] is a preliminary 4-page paper reporting on the same project, published in the ICML workshop on principled approaches to deep learning. The existence of this workshop paper was mentioned in our original submission under footnote 0. Please note that the ICLR dual submission policy explicitly allows publishing articles that have previously appeared in workshops (https://iclr.cc/Conferences/2018/CallForPapers).
>
> [1] T.S. Cohen, M. Geiger, J. Koehler, M. Welling, Convolutional Networks for Spherical Signals. In Principled Approaches to Deep Learning Workshop ICML 2017.

---

### Public Comment · (anonymous) · 2018-04-19
**Equivariance under non-linearity**

The paper nicely and theoretically propose an equivariant spherical cross-correlation for the rotation group. But it is not clear how the equivariance maintains in multiple layers with ReLU and BN inserted in between as the authors did in the experiments?

Sec 5.1 also shows that adding ReLU increase the difference by a large magnitude.

---

> ### Author Response · Authors · 2018-04-21
> **Good point**
>
> This is a good point. The network is equivariant if all the layers are equivariant, so that is what we must show. It was shown in the paper "Group Equivariant Networks" (section 6.2) that arbitrary pointwise nonlinearities are equivariant to the action of the group. This is true for the so-called regular representations, which act by permuting the neurons, whereas other (steerable / induced) representations may require special equivariant nonlinearities.
>
> The regular representation is what we denote by L_R in this paper:
> L_R f = f R^{-1},
> where juxtaposition means composition. Applying a pointwise nonlinearity s to a feature map f can be written mathematically as:
> C_s f = s f
>
> Since L_R acts by composing on the right and C_s acts by composing from the left, we have:
> L_R C_s f = L_R (s f) = (s f) R^{-1} = s (f R^{-1}) = C_s L_R f.
> That is, the regular representation L_R and the nonlinear operator C_s commute.
>
> This is the continuous theory. In practice, the numerical implementation results in a tiny loss of equivariance per linear layer (Fig. 3, top right). When ReLUs are used between each layer, we see in Fig 3. bottom right that the error is substantially larger, but does not increase meaningfully with depth. The reason for this is as follows: in order to measure the equivariance error Delta, we have to rotate the feature maps. Rotation of feature maps is exact (up to floating point error) only for band-limited signals, but the ReLU will introduce many high-frequency signals that cannot be exactly rotated with sub-pixel precision. So as soon as we use one layer of ReLU's, the error jumps. However, this appears to be an artefact of the measurement procedure (the numerical rotation step, to be precise) and does not seem to get worse with depth. This is mentioned in the last section before section 5.2: "This indicates that the error is not due to the network layers, but due to the feature map rotation, which is exact only for bandlimited functions".
>
> Batch normalization is exactly equivariant, as long as one uses one mean and std per feature map on SO(3). This is because both the mean and std are "scalars" in the geometrical sense that they are invariant under rotation. So we can multiply by them without affecting the equivariance.
>
> Beyond the equivariance error (Delta) experiments, the generalization results for spherical MNIST provide further support for the numerical accuracy of our implementation. If the numerical problems were severe, we would not expect to see such good generalization from a non-rotated training set to a rotated test set.
>
> In my ICLR talk, I will show a figure showing the feature maps for a rotated and non-rotated input. This allows you to easily see that the network is properly equivariant.

---

### Decision · Program_Chairs · 2018-01-29
**ICLR 2018 Conference Acceptance Decision**

**Decision:**

Accept (Oral)

**Comment:**

This work introduces a trainable signal representation for spherical signals (functions defined in the sphere) which are rotationally equivariant by design, by extending CNNs to the corresponding group SO(3). The method is implemented efficiently using fast Fourier transforms on the sphere and illustrated with compelling tasks such as 3d shape recognition and molecular energy prediction.

Reviewers agreed this is a solid, well-written paper, which demonstrates the usefulness of group invariance/equivariance beyond the standard Euclidean translation group in real-world scenarios. It will be a great addition to the conference.